# Relative abrasive potential of silica-based conventional and tablet dentifrices on enamel and dentin

**Shaira R. Kasi[1], Rafaela Oliveira Pilecco[2], Lucas Saldanha da Rosa[3], João Paulo Mendes Tribst[1]\*, Cornelis Johannes Kleverlaan[4], Mutlu Özcan[5], Albert J. Feilzer[1,4]**

**1** Department of Restorative and Reconstructive Oral Care, Academic Centre for Dentistry Amsterdam (ACTA), Universiteit van Amsterdam and Vrije Universiteit Amsterdam, Amsterdam, The Netherlands, **2** Department of Conservative Dentistry, Faculty of Dentistry, Federal University of Rio Grande do Sul, Porto Alegre, Brazil, **3** Post-Graduate Program in Oral Sciences (Prosthodontics Units), Faculty of Odontology, Federal University of Santa Maria (UFSM), Santa Maria, Brazil, **4** Department of Dental Materials Science, Academic Centre for Dentistry Amsterdam (ACTA), Universiteit van Amsterdam and Vrije Universiteit Amsterdam, Amsterdam, The Netherlands, **5** Center for Dental Medicine, Clinic for Masticatory Disorders and Dental Biomaterials, University of Zurich, Zurich, Switzerland

\* j.p.mendes.tribst@acta.nl

**Data Availability Statement:** All data generated or analyzed during this study are included in this published article. No additional data are available outside of what is presented within the paper.

## Abstract

### Objectives

This in vitro study aimed to investigate the toothbrushing wear on both enamel and dentin surfaces of reference and commercially available dentifrices.

### Methods

Bovine enamel and dentin blocks were initially polished and embedded within a resin composite in square shapes ($10 \times 8 \times 6$ mm$^3$). Employing toothbrushes equipped with nylon bristles, a toothbrushing machine was utilized, subjecting dentin specimens (n = 36) to 500 brush cycles and enamel samples to 5000 brush cycles (n = 36). Before and after the brushing simulation (2.45-N, 180 strokes per minute), an advanced contact profilometer was employed to measure the abraded depth. The wear rates were analyzed by using One-way ANOVA with two-sided Dunnett's multiple comparisons with a control (water).

### Results

Significant variations were observed among the tested toothpaste formulations, particularly in dentin wear, where Sident and Prodent showed notably higher values (7.30 µm and 9.67 µm, respectively) compared to the water control group (0.79 µm). Prodent also induced the highest enamel wear (2.64 µm) among the toothpaste formulations, while water, Zendium, and Denttabs exhibited comparatively lower enamel wear values. Statistical analysis using One-way ANOVA with two-sided Dunnett's multiple comparisons against the control (water) confirmed these differences.

**Funding:** The author(s) received no specific funding for this work.

**Competing interests:** The authors have declared that no competing interests exist.

## Significance

Toothbrushing with water causes minimal wear on enamel and dentin tissues, suggesting the predominant effect of three-body wear when using an abrasive medium. Comparing the standard references for dentifrice abrasives, Sident and Sylodent exhibit similar wear rates, making them reliable choices for in-vitro tests. When employed in a similar frequency, the wear rate of commercial toothpaste depends more on its composition than its form (paste or tablets).

## Introduction

For over a century toothbrushing with fluoridated toothpaste has been the standard for people's oral hygiene [1]. However, there is a growing number of people who reject the use of toothpaste because of the tubes that might contain (micro-)plastics affecting the environment [2]. Annually, more than a billion toothpaste tubes end up in landfills or oceans, which creates a massive amount of micro-plastic waste [3]. This is one of the reasons why a growing number of people are opting for toothpaste tablets as an alternative due to their environmental benefits, including reduced plastic waste and portability.

In summary, the components of manufacturing dentifrice (paste or tablets) can be divided into two categories: formulation excipients and active ingredients. Besides the 'formulation excipients' that are the base of making a paste or tablet 'active ingredients' are added that aim to benefit the therapeutic effects of the dentifrice. However, the effect of different vehicles during dentifrice usage on the wear rate of hard dental tissues is lacking.

An ISO standard (NEN-ISO 11609) [4] is available for assessing dentifrices, outlining requirements, test methods, and manufacturing conventional toothpaste. This standard is commonly utilized to define the quality of toothpaste necessary for conformity assessment of CE marking (Conformité Européenne) or FDA ('Food and Drug Administration', USA) approval. However, it does not provide guidelines for testing toothpaste tablets. Additionally, one of the standard reference abrasives is a silica-based powder (Sident, Evonik, Wolfgang, Germany) that is not available on the market anymore. However, an alternative abrasive with similar components, named Sylodent, was developed by the same manufacturer, which has not been previously evaluated concerning its abrasive properties.

With the aid of standardized testing methods (ISO), the abrasive potential of dentifrices can be assessed in vitro. This is particularly important in dental wear as toothbrushing has been identified as one important factor that might accelerate the process, especially in cervical areas. The level of abrasiveness varies significantly among dentifrices and therefore their potential to contribute to dental wear [5].

In essence, the wear process during brushing methods consists of three-body wear. This means that this process is not only the interaction between two contacting surfaces (toothbrush filaments and enamel or dentin) but also the impact of intervening particles or substances acting as a third body in between the surfaces. In this sense, the comparison with brushing only with water could demonstrate if the wear rate was enhanced by the slurry solution or only by the brush effect on the dental surface. Unfortunately, most studies do not apply water as a negative control, and neither does the ISO indicate that.

Despite the inherent limitations, toothbrushing simulation is a wear testing methodology considered instrumental for researchers and manufacturers in appraising the resilience and

efficacy of dental materials under conditions that closely mimic real-world scenarios [6, 7]. This study aimed to assess and compare tooth wear caused by water, classic and new silica-reference slurries as well as conventional toothpaste and toothpaste tablets in enamel and dentin tissues. The null hypothesis was that the wear rate would be similar between different abrasive solutions regardless of the dental tissue that is being evaluated [7].

## Materials and methods

### Study design

The materials tested in this study were two toothpastes, two toothpaste tablets, and two reference pastes made according to the instructions of ISO 11609:2017.

The ISO standard was used as the basis for the requirements, test methods, and markings. It describes two methods for testing the abrasiveness of dentifrices. In this study, contact profilometry was used for the determination of abraded depth after brushing. This method has been established to be equivalent to the radio-tracer method and is referred to as RDA-PE (Relative Dentin Abrasion–Profilometry Equivalent) and REA-PE (Relative Enamel Abrasion–Profilometry Equivalent).

The independent variables included the type of dentifrice (reference and commercially available), the type of dental substrate (enamel or dentin), and the brushing conditions (500 brush cycles for dentin and 5000 brush cycles for enamel). The dependent variable was the depth of abrasion measured in micrometers (μm) before and after brushing using a contact profilometer.

### Reference toothpaste

According to the ISO, it is necessary to compare a reference abrasive when evaluating the wear rate of toothpaste. The described reference toothpaste in the ISO standard is based on calcium pyrophosphate or silica. When using a silica-based reference abrasive, the material indicated is Sident® (Evonik, Hanau, Germany). Evonik no longer manufactures Sident®, and the company has provided a replacement abrasive named Sylodent®. In this study, both materials were used to compare them and evaluate whether their wear rates are comparable as a reference abrasive for toothbrushing tests.

The liquid used with both silica-based abrasives was the reference diluent containing 0,5% carboxymethylcellulose sodium salt solution in 10% glycerin. To prepare 1 liter of the diluent, 50 ml of glycerine was heated to 60°C and added 5 g of Carboxymethylcellulose while stirring. When the mixture was homogeneous, another 50 ml of heated glycerin was added and the solution was mixed for another 60 minutes. After this step, 900 ml of distilled water was added and allowed to cool while stirred overnight. The viscosity was stabilized by allowing the solution to stand overnight before usage. The reference diluent was then used following the recommended concentration to make up slurries of the reference abrasives (Sident® and Sylodent®). Both solutions were stored for subsequent utilization.

### Toothpaste and tablets

According to the ISO standard, 25 grams of toothpaste should be diluted with artificial saliva or deionized water. The standard does not describe testing toothpaste tablets. While tablets do have a constant weight per use, the amount of toothpaste can differ per person and toothbrush size, leading to the question of how much tablets are comparable to 25-gram toothpaste to calculate a comparable ratio from toothpaste powders to create a testing slurry that matches that of toothpaste. The Dutch Scientific Association Ivory Cross (Ivoren Kruis) advises adding

**Table 1. Selected tubes of toothpaste, tablets their manufacturers, and chemical compositions.**

| | Paste | Manufacturer | | Composition |
|---|---|---|---|---|
| 1 | Reference toothpaste | Grace GmbH. Worms, Germany, In der Hollerhecke | Sident[a] | Silica, sodium |
| 2 | Alternative reference toothpaste | Grace GmbH. Worms, Germany, In der Hollerhecke | Sylodent[a] | |
| 3 | Zendium[a] Classic | Unilever. London, United Kingdom: Unilever House, 100 Victoria Embankment, London, EC4Y 0DY. | Toothpaste | Aqua, Hydrated Silica, Sorbitol, Glycerin, Steareth-30, Xanthan Gum, Aroma, Carrageenan, Disodium Phosphate, Sodium Fluoride, Amyloglucosidase, Citric Acid, Zinc Gluconate, Sodium Benzoate, Glucose Oxidase, Sodium Saccharin, Potassium Thiocyanate, Lysozyme, Colostrum, Lactoferrin, Lactoperoxidase, CI 77891 |
| 4 | Prodent Coolmint | Unilever. London, United Kingdom: Unilever House, 100 Victoria Embankment, London, EC4Y 0DY. | Toothpaste | Aqua, Hydrogenated Starch Hydrolysate, Hydrated Silica, PEG-32, Sodium Lauryl Sulfate, Aroma, Cellulose Gum, Benzyl Alcohol, Sodium Fluoride, Sodium Saccharin, Limonene, Eugenol, CI 77891. |
| 5 | Smyle Brush Mint with Fluoride | Smyle, Hannie Dankbaarpassage, Amsterdam, The Netherlands | Toothpaste tablet | Sorbitol, yeast, hydroxyapatite, acacia senegal gum extract, sodium coco, sulfate, hydrated silica, mentha arvensis leaf oil, sodium, monofluorophosphate (fluoride), magnesium stearate, limonene, mentha viridis leaf oil, stevioside, menthol, mentha piperita oil, linalool, calcium carbonate. |
| 6 | Oceonics Toothpaste Denttabs with fluoride | Denttabs, 13347 Berlin, Germany | Toothpaste tablet | Microcrystalline cellulose, sodium bicarbonate, citric acid, silica, sodium lauroyl glutamate, magnesium stearate, natural mint aroma, menthol, stevia, xanthan gum, sodium fluoride, eugenol. |

approximately 1.5 cm of toothpaste to the toothbrush, which corresponds to 1.45 grams. Thus, 25 grams of toothpaste correspond to 17 toothbrushing sessions. With this knowledge, we took 17 tablets for the test with Smyle Brush Mint with Fluoride and 17 tablets for Oceonics. The 17 tablets were dissolved in 40 ml water as the ISO prescribed 25 grams of toothpaste to be diluted in 40 ml water. Two reference kinds of toothpaste, two brands of toothpaste, and two brands of toothpaste tablets were used (Table 1).

## Dentin and enamel test samples

Bovine enamel and dentin were used for all the tests. This study did not involve the use of live animals or require any procedures that would have affected animal welfare. The bovine teeth used in the research were obtained from a local butcher shop as by-products from animals already processed for commercial purposes. Since the study utilized discarded materials and did not involve live animal experimentation, no prospective approval from an Institutional Animal Care and Use Committee (IACUC) or other relevant ethics board was necessary. Additionally, there was no need for anesthesia, euthanasia, or any other form of animal sacrifice as part of this study.

After collecting the bovine teeth, each tooth crown was separated from the root using a handpiece and constant water cooling. Each bovine crown was then sectioned using a precision cutting machine to obtain enamel samples (max 4 mm x 4 mm) from the labial surfaces of 72 nondamaged bovine incisors, which were stored in refrigerated water until use. The enamel surfaces were ground flat and polished with water-cooled SIC sandpaper (400 grit; Buehler CarbiMet, Germany), thereby removing approximately 100–150 μm of the outermost enamel layer. The samples were embedded in a plexiglass baseplate of the brushing machine with resin composite material (3M™ Filtek™ Z250 Universal Restorative). The sample surfaces were subsequently exposed to 30 μm higher than the baseplate level and polished with water-cooled SIC sandpapers (1200 grit; Buehler CarbiMet, Germany). For the dentin specimen, the same procedure was applied with as difference that in the first step, a great part of the surface was polished until exposing the dentin. In total 18 specimens were used per substrate.

Every baseplate contained six samples of dentin or enamel. Similar to previous studies, each specimen was embedded leaving 2.0 mm of exposed material from the baseplate. In sequence, a cover with a thin metal foil was placed on top of it to standardize 3–5 mm of surface in contact with the toothbrush. The specimens were hydrated during all preparation, abrasion, and measurement procedures.

## Toothbrushing simulation

On the baseplate, a metal frame was fixed to create six chambers where the toothpaste was filled during the test. The toothbrushes (iQ-Hard Lactona®) were positioned above the mounted specimens with a designated tension on the brush (2.45 N) while immersed in a dentifrice slurry. The base plate moves back and forth with a sliding action of 1 cm at 2Hz [7]. Before the first time, the brushes were pre-conditioned for 20,000 strokes on the brushing machine in water under standard conditions. The brushes were frequently inspected for wear and changed for each baseplate. To reduce the biological variation of the enamel and dentin samples, each tested solution was used in each chamber sequentially, and the wear rate was calculated as the difference between profile measurements from previous and new values.

For dentin, the specimens were brushed for 500 strokes in all the tests. Enamel is significantly harder than dentin and the depth of abrasion for enamel will be significantly lower versus that of dentin [8]. To ensure a proper dynamic range of mean depth of abrasion, enamel specimens were brushed for 5000 strokes, according to the ISO.

## Profilometer measurement

A profilometer with a sensitivity of $<0,1$ µm was used to determine the abraded depth after brushing (SJ-400, Mitutoyo, Kawasaki, Japan). The mean depth of abrasion was measured as abraded depth versus the control masked area on the specimen surface. The profiles started 0.5 mm from the previously taped (controlled) area of the specimen across the abraded zone and into the opposite previously taped (control) area opposite at least 0.5 mm. The total length of the trace was 4 mm, 0.5 mm on one side of the abraded area, 3 mm on the abraded area, and 0.5 mm on the opposite side of the abraded area. The tracing was done for 2 traces evenly spaced across the abraded area, avoiding the boundary. Every trace resulted in a measurement of 4.000 points that were imported into Microsoft Excel 2019 software (Microsoft, Inc., Redmond, Washington). First, the traces were leveled out by bringing the unworn references to zero level, whereafter the mean depth for every trace of abrasion was determined. Fig 1 shows a representative wear measurement in the blue line after post-processing.

**Statistical analysis.** The data was analyzed in statistical software (SPSS v.21, IBM) with a significance level of 0.05. One-way ANOVA with two-sided Dunnett's multiple comparisons was used to show which and if the results of the toothpaste significantly differed from the control (water).

## Results

In total, 144 wear measurements were obtained; 72 for dentin and 72 results for enamel. The mean values and standard deviations (SD) of tooth wear, both in dentin and enamel, resulting from the use of different toothpastes are summarized in Tables 2 and 3.

The tooth wear measurements, expressed in micrometers (µm), revealed notable variations among the tested toothpaste formulations (Fig 2). Sident and Prodent exhibited significantly higher dentin wear (7.30 µm and 9.67 µm, respectively) compared to the control group treated with water (0.79 µm). Enamel wear also demonstrated variations, with Prodent inducing the highest enamel wear (2.64 µm) among the toothpaste formulations (Fig 3). In contrast, water, Zendium, and

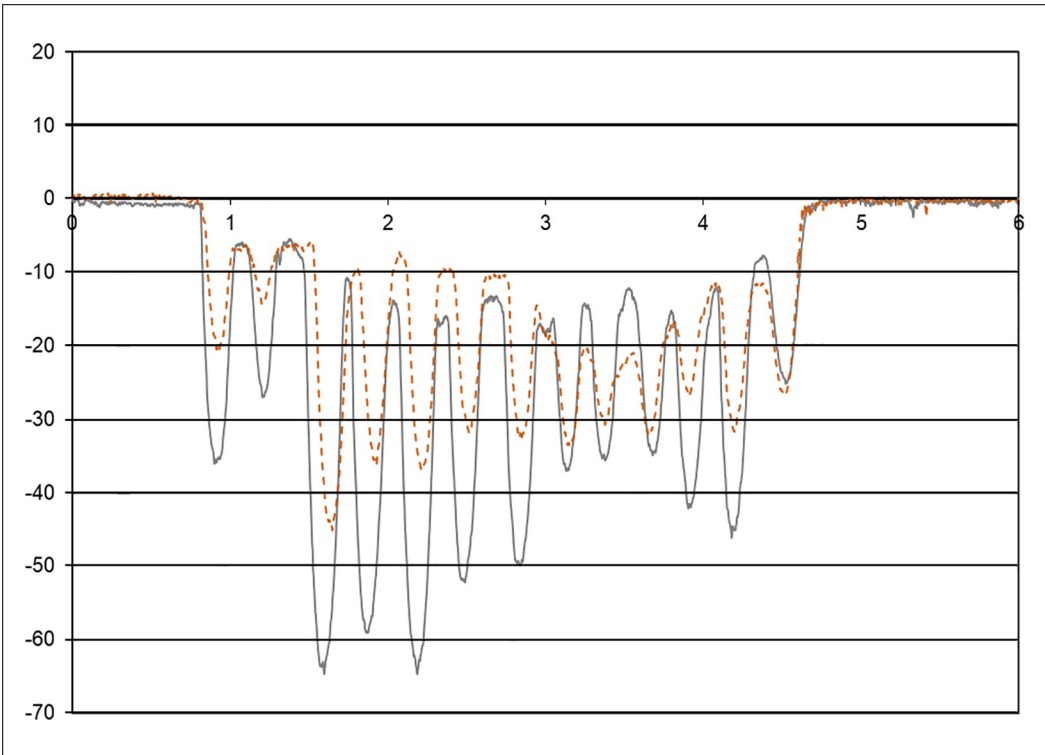

**Fig 1. Representative images of the tooth wear measurements in between tooth brushing cycles–calculated between the difference of the black line and the orange line.**

Denttabs exhibited comparatively lower enamel wear values. Statistical analysis using One-way ANOVA with two-sided Dunnett's multiple comparisons against the control (water) highlighted these observed differences, as indicated by distinct superscript letters in each column.

A one-way Analysis of Variance (ANOVA) was conducted to evaluate the differences in Dentine measurements between groups. The ANOVA results indicated a statistically significant effect, with a p-value of 0.01 and an F-statistic of 3.69. To further investigate which specific group means were significantly different, the Tukey test was performed with an alpha level set at 0.05 (Table 4).

The results indicated that Prodent was significantly different from Sident, Smyle, Denttabs, and Zendium; Sident was significantly different from Prodent and Zendium; Sylodent was significantly different from Prodent; Smyle was significantly different from Prodent; Denttabs was significantly different from Prodent, Sident, and Sylodent; and Zendium was significantly different from Prodent and Sident. These results suggest that the choice of toothpaste brand has a significant impact on the measured outcome, with specific brands showing distinct differences from others.

In addition, a one-way Analysis of Variance (ANOVA) was performed to assess the differences in Enamel measurements across various groups. The results revealed no significant differences among the groups, as indicated by a p-value of 0.25 and an F-statistic of 1.40.

## Discussion

The null hypothesis, positing similar wear rates between reference pastes with different standard abrasives, was refuted, emphasizing the significance of specific toothpaste formulations

**Table 2. Measured values per point after contact profilometer assessment.**

| Dentine | Sident | Sylodent | Zendium | Prodent | Smyle | Denttabs | Water |
|---|---|---|---|---|---|---|---|
| | 2.291 | 4.008 | -0.760 | 12.008 | 4.028 | 1.570 | -0.137 |
| | 6.320 | 4.052 | -3.851 | 7.417 | 7.321 | -0.381 | -0.065 |
| | 6.007 | 4.027 | 2.511 | 4.807 | -0.805 | 7.172 | -6.932 |
| | 16.239 | 14.348 | 2.431 | 11.076 | 7.908 | 3.398 | 1.244 |
| | 3.224 | 2.644 | 2.108 | 4.507 | 1.217 | 0.139 | -0.206 |
| | 9.746 | 8.790 | 2.256 | 18.184 | 6.761 | 4.592 | 1.347 |
| Enamel | Sident | Sylodent | Zendium | Prodent | Smyle | Denttabs | Water |
| | 2.537 | 1.528 | -0.006 | 1.751 | 0.548 | 0.072 | 0.114 |
| | -0.458 | -2.169 | 4.924 | 3.022 | 0.811 | 0.384 | 0.209 |
| | 4.139 | 2.729 | 0.544 | 6.189 | 5.720 | 1.607 | -1.584 |
| | 1.963 | 0.379 | 0.408 | 1.546 | 6.250 | -1.802 | -0.858 |
| | -0.326 | 1.866 | -0.106 | 0.625 | 0.060 | 0.035 | 0.102 |
| | 0.461 | 1.263 | 0.420 | 2.723 | 1.149 | 0.776 | -0.353 |

in influencing tooth wear. These findings provide valuable insights into dental abrasion and the effects of various dentifrices. Notably, brushing with water had minimal impact, highlighting the pivotal role of dentifrice composition in tooth wear, which surpasses the influence of brushing action. While this underscores the importance of dentifrices for effective dental cleansing, it also emphasizes the need for careful selection to optimize dental care [7]. In addition, the negative values for the water group indicate minimal surface loss, as expected from a non-abrasive control.

When comparing Sident and Silodent, both indicated as silica references for laboratory dentifrice assessment methods, no difference was observed for either enamel or dentin tissue. This aligns with the manufacturer's indication, suggesting that the ISO should be updated accordingly. Additionally, the study revealed that dentin exhibited higher wear rates compared to enamel under toothbrushing conditions, emphasizing the importance of caution when brushing the cervical region of the tooth [9].

The study further elucidated that the composition of the dentifrice exerts a more pronounced influence on tooth wear than the delivery vehicle. Enamel, being the outermost and hardest layer of the tooth, is resilient to abrasion, whereas dentin, being softer and more porous, is more susceptible to wear. Comparing traditional toothpaste formulations with emerging alternatives, such as toothpaste tablets, is crucial amidst increasing environmental concerns associated with toothpaste tube waste [10–12].

**Table 3. Mean and standard deviation (SD) of tooth wear (dentin and enamel) in µm\*.**

| Toothpaste | Dentin wear (SD) | Enamel wear (SD) |
|---|---|---|
| Water | -0.79 (3.09)[A] | -0.39 (0.71)[A] |
| Sident | 7.30 (5.10)[B] | 1.39 (1.81)[A] |
| Sylodent | 6.31 (4.47)[B] | 0.93 (1.70)[A] |
| Zendium | 0.78 (2.59)[A] | 1.03 (1.92)[A] |
| Prodent | 9.67 (5.20)[B] | 2.64 (1.94)[B] |
| Smyle | 4.41 (3.57)[A] | 2.42 (2.79)[A] |
| Denttabs | 2.75 (2.88)[A] | 0.18 (1.13)[A] |

\*Different superscript letters in each column indicate statistical differences according to One-way ANOVA with two-sided Dunnett's multiple comparisons with a control (water)

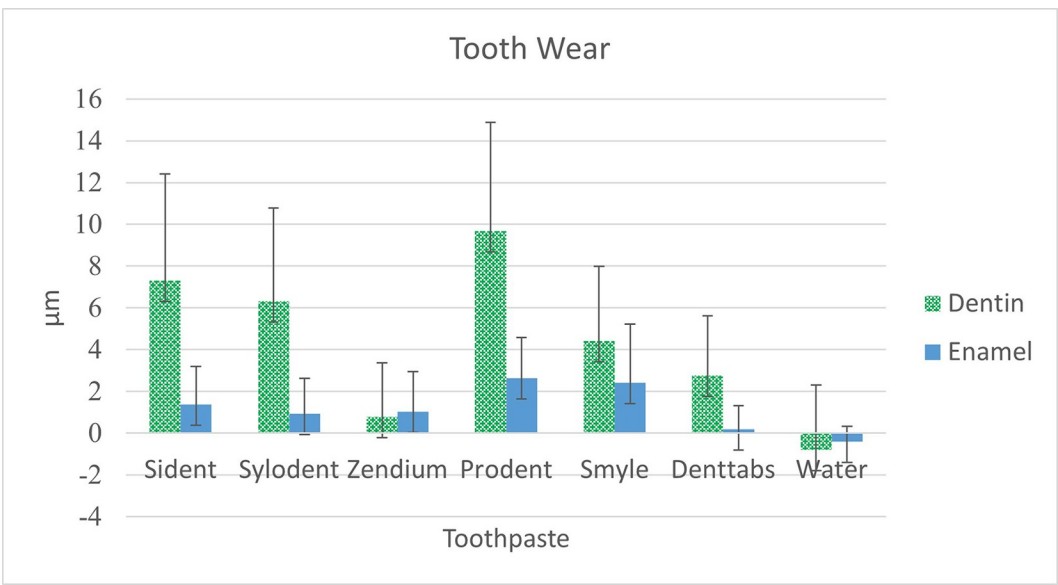

**Fig 2. Bar graph showing the average wear per group.**

Toothpaste tablets claimed as a sustainable oral care solution, exhibited comparable wear rates to conventional toothpaste. Previous studies have highlighted their benefits in maintaining the gloss and surface roughness of resin-based composite materials. However, their environmental impact must be considered holistically, taking into account manufacturing, consumption, and disposal practices [13, 14]. The tablet composition, including factors such as the use of different binders and the physical structure of the abrasive particles in tablet form,

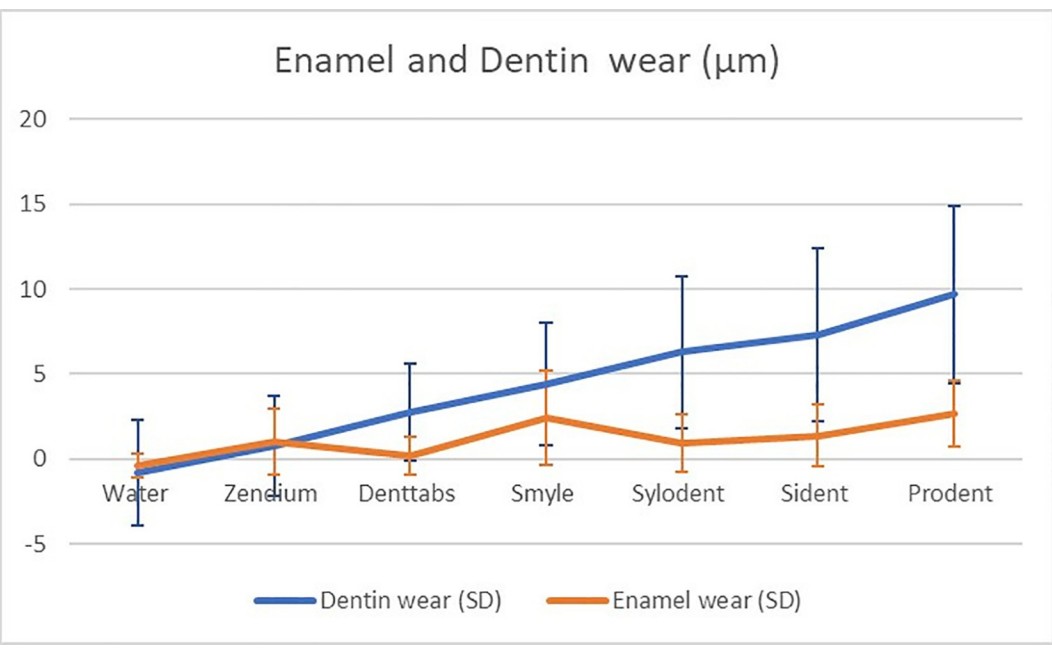

**Fig 3. Graphical plot of the mean of wear (in μm) for each group in dentin (blue) and enamel (orange) in ascending order for dentin.**

**Table 4. Mean and standard deviation (SD) of tooth wear (dentin and enamel) in μm\*.**

| Group 1 | Group 2 | Mean-diff | P-Adj | Lower | Upper | Reject |
|---|---|---|---|---|---|---|
| Denttabs | Prodent | 7.25 | 0.001 | 3.96 | 1.054 | TRUE |
| | Sident | 5.22 | 0.001 | 1.93 | 8.51 | TRUE |
| | Smyle | 2.32 | 0.268 | -0.96 | 5.61 | FALSE |
| | Sylodent | 4.56 | 0.001 | 1.271 | 7.85 | TRUE |
| | Zendium | 1.96 | 0.401 | -1.32 | 5.25 | FALSE |
| Prodent | Sident | -2.02 | 0.384 | -5.32 | 1.26 | FALSE |
| | Smyle | -4.92 | 0001 | -8.22 | -1.63 | TRUE |
| | Sylodent | -2.68 | 0.147 | -5.98 | 0.60 | FALSE |
| | Zendium | -5.28 | 0.001 | -8.57 | -1.99 | TRUE |
| Sident | Smyle | -2.90 | 0.108 | -6.19 | 0.39 | FALSE |
| | Sylodent | -0.66 | 0.999 | -3.95 | 2.61 | FALSE |
| | Zendium | -3.25 | 0.048 | -6.54 | -0.03 | TRUE |
| Smyle | Sylodent | 2.24 | 0.232 | -1.05 | 5.53 | FALSE |
| | Zendium | -0.35 | 0.999 | -3.64 | 2.93 | FALSE |
| Sylodent | Zendium | -2.59 | 0.168 | -5.88 | 0.69 | FALSE |

may reduce or modify its interaction with enamel and dentin surfaces, potentially affecting its abrasive properties.

Considering the diversity in brushing behaviors and dentifrice concentrations, future studies should explore how variations in dentifrice concentration, influenced by individual brushing habits, may affect tooth wear. This necessitates a nuanced approach to tooth wear assessment to capture the full spectrum of oral care practices [15, 16].

An important issue highlighted is the absence of an ISO standard for testing toothpowders or tablets. While the present adjusted method offers a potential workflow for testing toothpowders and tablets, further refinement and validation are required. In summary, this study enhances our understanding of tooth wear dynamics and prompts critical considerations for the dental community and consumers. Dentifrice composition emerges as a key determinant of tooth wear, advocating for formulations that prioritize efficacy and environmental sustainability as oral care products continue to evolve [17].

While our study provides valuable insights, it's important to acknowledge its limitations. In the oral environment, several factors beyond toothbrushing, such as pH levels, temperature fluctuations, and dietary habits, can influence the interface and wear characteristics of restorative materials [17]. Nonetheless, understanding the mechanical effects of toothbrushing remains essential, particularly when dealing with diverse tissues and/or restorative materials. In addition, we selected 5,000 strokes based on established protocols. However, we acknowledge that other factors, such as the abrasiveness of the dentifrices and the simulation setup, could influence the wear outcome.

## Conclusion

Based on the findings of this in vitro study, the following conclusions were drawn:

1. The observed variations in dentin and enamel wear among the tested formulations show the importance of considering the impact of toothpaste ingredients on dental tissues;

2. Dentifrices play a pivotal role in toothbrushing wear since brushing only with water promoted insignificant wear rates on both enamel and dentin tissues;

3. Sylodent is a valid alternative to the no-longer-manufactured Sident and presents wear rates adequate to be used as Silica-based reference slurry;

4. Toothpaste tablets are comparable to conventional toothpaste, with some specific formulations, such as Prodent cream, when considering the amount of wear caused on dental hard tissues.

## Author Contributions

**Conceptualization:** Shaira R. Kasi, João Paulo Mendes Tribst, Cornelis Johannes Kleverlaan, Mutlu Özcan, Albert J. Feilzer.

**Data curation:** Shaira R. Kasi, Rafaela Oliveira Pilecco, Lucas Saldanha da Rosa, Albert J. Feilzer.

**Formal analysis:** Shaira R. Kasi, Rafaela Oliveira Pilecco, Lucas Saldanha da Rosa.

**Funding acquisition:** Cornelis Johannes Kleverlaan, Albert J. Feilzer.

**Investigation:** Shaira R. Kasi, Rafaela Oliveira Pilecco, Lucas Saldanha da Rosa, João Paulo Mendes Tribst, Cornelis Johannes Kleverlaan, Mutlu Özcan, Albert J. Feilzer.

**Methodology:** Shaira R. Kasi, Rafaela Oliveira Pilecco, Lucas Saldanha da Rosa, João Paulo Mendes Tribst, Albert J. Feilzer.

**Project administration:** João Paulo Mendes Tribst, Mutlu Özcan, Albert J. Feilzer.

**Resources:** João Paulo Mendes Tribst, Cornelis Johannes Kleverlaan, Mutlu Özcan, Albert J. Feilzer.

**Software:** Mutlu Özcan.

**Supervision:** João Paulo Mendes Tribst, Cornelis Johannes Kleverlaan, Mutlu Özcan, Albert J. Feilzer.

**Validation:** Cornelis Johannes Kleverlaan, Albert J. Feilzer.

**Visualization:** João Paulo Mendes Tribst, Cornelis Johannes Kleverlaan.

**Writing – original draft:** Shaira R. Kasi, Rafaela Oliveira Pilecco, Lucas Saldanha da Rosa, João Paulo Mendes Tribst.

**Writing – review & editing:** Shaira R. Kasi, João Paulo Mendes Tribst, Cornelis Johannes Kleverlaan, Mutlu Özcan, Albert J. Feilzer.

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
