## [Decision Letter · Decision Letter 0]

3 Sep 2024

PONE-D-24-31778Influence of dentifrices on toothbrushing wear of enamel and dentinPLOS ONE

Dear Dr. Tribst,

Thank you for submitting your manuscript to PLOS ONE. After careful consideration, we feel that it has merit but does not fully meet PLOS ONE’s publication criteria as it currently stands. Therefore, we invite you to submit a revised version of the manuscript that addresses the points raised during the review process.

We look forward to receiving your revised manuscript.

Kind regards,

Geelsu Hwang, Ph.D.

Academic Editor

PLOS ONE

Journal Requirements:

Reviewers' comments:

Reviewer's Responses to Questions

**Comments to the Author**

1. Is the manuscript technically sound, and do the data support the conclusions?

Reviewer #1: Yes

Reviewer #2: Yes

Reviewer #3: Partly

2. Has the statistical analysis been performed appropriately and rigorously? 

Reviewer #1: Yes

Reviewer #2: Yes

Reviewer #3: Yes

3. Have the authors made all data underlying the findings in their manuscript fully available?

Reviewer #1: Yes

Reviewer #2: Yes

Reviewer #3: Yes

4. Is the manuscript presented in an intelligible fashion and written in standard English?

Reviewer #1: Yes

Reviewer #2: Yes

Reviewer #3: Yes

5. Review Comments to the Author

Reviewer #1: The topic evaluated in this article is very interesting and of clinical relevance. The evaluation of dentifrices as tablets is very new in literature, and on eof the novelty of the study. This should be highlighted in the title. Here is a suggestion: Abrasive Potential of Silica-Based Conventional and Tablet Dentifrices on Enamel and Dentin.

In the discussion and conclusion there are statmets comparing the dentifrices in the conventional form and the tablets. However, the table 2 shows the results of Dunnett test, comparing them with the control (water) and Figure 2 and 3 does not present and statistical result regarding the comparison between groups. And additional analysis, such as Tukey test, would be interesting to supports those findings. Still, Prodent, for example, seems to present a higher surface loss in dentin than the tablets. Was that thrue? Is so, the concluison number 4 should be re-written. Also, the first sentence in the 4th paragraph of discussion needs clarification regarding this topic. It seems that the composition of the tablet can interfere with its abrasive power.

Wghy the authors opt to perform 500 cycles in dentin and 5000 in enamel? was this suggested by the ISO? This was not clear for me, while reading.

Reviewer #2: The study investigated tooth surface wear caused by brushing with different commercial toothpastes on enamel and dentin. Although the study is well written and the methodology is sound, some toothpastes have a local circulation, meaning that they are not sold in every country, which limits the relevance of the study. In addition, it is already known that toothpastes have different abrasively, thus, the rationale behind the study needs to be better explained. If the idea is to evaluate the abrasive effect of pellets and compared them with toothpastes, this should be clear in the title, abstract and introduction.

My specific comments follow below:

Abstract

-It is not clear how many toothpastes were tested and what was the profilometer used (contact or optical). Please insert this info to the text.

Introduction

-By the introduction, it seems that the goal of the study is to evaluate the viability of measuring the abrasiveness of toothbrushing pallets with de REA-PE and RDA-PE methods. If that is so, this should be reflected in the title and abstract.

Materials and Methods

-Please clarify how many specimens per group were used.

Results

-Table 2: Surface loss for the water group appeared negative, what this means?

Reviewer #3: The manuscript reports an interesting topic, in which the enamel and dentin wear was assessed with reference and commercially available dentifrices. However, the study presents some issues that need to be clarified, mainly with the model and methodology adopted.

Considerations:

Abstract:

1) Methods: is the “toothbrushing wear testing device” a toothbrushing machine? Can it not be named as “toothbrushing machine”? The authors repeated “brush” before cycles, but it is obvious that the cycles comprised brushing. What is an “advanced profilometer”? Optical? Contact?

2) Results: the last phrase is not necessary since the authors reported the significant difference with water in the third phrase.

Material and Methods:

1) Study design: the authors did not describe some essential characteristics of a study design. Please provide the factors and specify the dependent variable and the experimental units. The justification for using the profilometer should be inserted in the “profilometer measurement”.

2) Reference toothpaste: the authors did not provide the meaning of CMC before using it. Please add it after “carboxymethylcellulose”. Did the authors measure the viscosity? Would it be comparable to the other dentifrices used?

3) How did the authors determine the appropriate sample size used?

4) How were the specimens stored before the experiments and after them?

5) The authors should describe the specimen’s preparation and the experiment with the toothbrushing in different sessions.

6) There are some values written with “comma”. Please replace them with “dot”.

7) How did the authors determine the number of strokes to be performed? Do 500 or 5000 strokes represent a clinical use? What type of toothbrush was used?

8) Could the authors inform the reproducibility of the method adopted as well as the coefficient of variation? In addition, did the authors check the initial specimen curvature?

9) Statistical analysis: the first line informing how many measurements were performed is not appropriate to this item as well as Table 2. This information should be described in “results”. Was the normality test performed?

Results

1) If the reference toothpastes did not promote enamel wear, would it be possible that the number of strokes was not enough and then the model should be reviewed?

6. PLOS authors have the option to publish the peer review history of their article (what does this mean?). If published, this will include your full peer review and any attached files.

Reviewer #1: No

Reviewer #2: No

Reviewer #3: No

---

## [Author Response · Author response to Decision Letter 0]

8 Oct 2024

Reviewer #1: The topic evaluated in this article is very interesting and of clinical relevance. The evaluation of dentifrices as tablets is very new in literature, and on eof the novelty of the study. This should be highlighted in the title. Here is a suggestion: Abrasive Potential of Silica-Based Conventional and Tablet Dentifrices on Enamel and Dentin. 

Answer: Dear reviewer, thank you for this suggestion. The title has been modified according to your and other reviewer’s comments. 

In the discussion and conclusion there are statmets comparing the dentifrices in the conventional form and the tablets. However, the table 2 shows the results of Dunnett test, comparing them with the control (water) and Figure 2 and 3 does not present and statistical result regarding the comparison between groups. And additional analysis, such as Tukey test, would be interesting to supports those findings. Still, Prodent, for example, seems to present a higher surface loss in dentin than the tablets. Was that thrue? Is so, the concluison number 4 should be re-written. Also, the first sentence in the 4th paragraph of discussion needs clarification regarding this topic. It seems that the composition of the tablet can interfere with its abrasive power. 

Answer: A new statistical test was conducted to compare the groups against one another. Additionally, the conclusion has been revised as per your suggestions. 

Wghy the authors opt to perform 500 cycles in dentin and 5000 in enamel? was this suggested by the ISO? This was not clear for me, while reading. 

Answer: Our initial plan was to conduct 5,000 strokes for both enamel and dentin. However, during the pilot study, due to the softer nature of dentin, we observed visible signs of wear after only 500 strokes. Given the significant difference in hardness between the two tissues, we decided to halt the test for dentin at this point, as it was already sufficient to detect the effects of each abrasive medium. 

Reviewer #2: The study investigated tooth surface wear caused by brushing with different commercial toothpastes on enamel and dentin. Although the study is well written and the methodology is sound, some toothpastes have a local circulation, meaning that they are not sold in every country, which limits the relevance of the study. In addition, it is already known that toothpastes have different abrasively, thus, the rationale behind the study needs to be better explained. If the idea is to evaluate the abrasive effect of pellets and compared them with toothpastes, this should be clear in the title, abstract and introduction. 

Answer: Thank you for your valuable feedback. We appreciate your acknowledgment of our study's writing and methodology. We recognized the limitation regarding the regional availability of some toothpaste. Additionally, we revised the title, abstract, and introduction to clearly articulate our objective of evaluating the abrasive effects of various toothpastes and pellets. 

My specific comments follow below: 

Abstract 

-It is not clear how many toothpastes were tested and what was the profilometer used (contact or optical). Please insert this info to the text. 

Answer: This information has been added. 

Introduction 

-By the introduction, it seems that the goal of the study is to evaluate the viability of measuring the abrasiveness of toothbrushing pallets with de REA-PE and RDA-PE methods. If that is so, this should be reflected in the title and abstract. 

Answer: The corrections have been performed. 

Materials and Methods 

-Please clarify how many specimens per group were used. 

Answer: We used 18 specimens per group. 

Results 

-Table 2: Surface loss for the water group appeared negative, what this means? 

Answer: The negative values for surface loss in the water group indicate that there was no measurable wear on the enamel or dentin surfaces after the brushing simulation. This result is consistent with our expectations, as water serves as a control and is not expected to cause any significant abrasion. 

Reviewer #3: The manuscript reports an interesting topic, in which the enamel and dentin wear was assessed with reference and commercially available dentifrices. However, the study presents some issues that need to be clarified, mainly with the model and methodology adopted. 

Answer: Thank you for your thoughtful review and for highlighting the relevance of our study on enamel and dentin wear with various dentifrices. We appreciate your feedback regarding the model and methodology. 

Considerations: 

Abstract: 

1) Methods: is the “toothbrushing wear testing device” a toothbrushing machine? Can it not be named as “toothbrushing machine”? The authors repeated “brush” before cycles, but it is obvious that the cycles comprised brushing. What is an “advanced profilometer”? Optical? Contact? 

Answer: The text has been corrected as suggested. 

2) Results: the last phrase is not necessary since the authors reported the significant difference with water in the third phrase. 

Answer: The text has been corrected as suggested. 

Material and Methods: 

1) Study design: the authors did not describe some essential characteristics of a study design. Please provide the factors and specify the dependent variable and the experimental units. The justification for using the profilometer should be inserted in the “profilometer measurement”. 

Answer: Thank you for your valuable feedback regarding our manuscript. In response to your comments on the study design, we have clarified the independent variables, including the type of dentifrice (reference and commercially available), the type of dental substrate (enamel or dentin), and the brushing conditions (500 brush cycles for dentin and 5000 brush cycles for enamel). We specified that the dependent variable is the depth of abrasion measured in micrometers (µm) and indicated the experimental units with sample sizes (n=36 for both enamel and dentin). 

2) Reference toothpaste: the authors did not provide the meaning of CMC before using it. Please add it after “carboxymethylcellulose”. Did the authors measure the viscosity? Would it be comparable to the other dentifrices used? 

Answer: It means Carboxymethylcellulose. It has been corrected in the text. We did not measure the viscosity since the amount of it was defined by the ISO. 

3) How did the authors determine the appropriate sample size used? 

Answer: The appropriate sample size was determined based on the results of a preliminary pilot study. 

4) How were the specimens stored before the experiments and after them? 

Answer: The specimens were stored in refrigerated water both before and after the experiments 

5) The authors should describe the specimen’s preparation and the experiment with the toothbrushing in different sessions. 

Answer: The methods have been updated with the specimen’s preparation and experiment in different sections. 

6) There are some values written with “comma”. Please replace them with “dot”. 

Answer: Thank you for bringing this to our attention. We have corrected the values to replace commas with dots as needed. 

7) How did the authors determine the number of strokes to be performed? Do 500 or 5000 strokes represent a clinical use? What type of toothbrush was used? 

Answer: In this study, we employed 500 strokes for dentin specimens and 5000 strokes for enamel specimens to reflect the differing wear characteristics and rates of abrasion experienced by these substrates under simulated toothbrushing conditions. Dentin is softer and more susceptible to wear compared to enamel. Therefore, a lower number of strokes (500) was used for dentin to accurately simulate the wear that would occur in a realistic brushing scenario without overly accelerating the wear process. Clinical toothbrushing varies significantly between individuals in terms of technique, pressure, and duration. Factors such as brushing speed, angle, and the specific motion used can greatly influence the effectiveness of plaque removal and the wear experienced by enamel and dentin. Thus, a standardized stroke count may not accurately represent real-world brushing habits. The toothbrush used was iQ-Hard Lactona. 

8) Could the authors inform the reproducibility of the method adopted as well as the coefficient of variation? In addition, did the authors check the initial specimen curvature? 

Answer: We appreciate your inquiry regarding the coefficient of variation. While we did not calculate the coefficient of variation for our measurements, it is important to note that our methodology is well-established and follows classical approaches used in the literature for similar studies. About the curvature of the samples, each specimen was manually flattened in a water-cooled polishing machine. 

9) Statistical analysis: the first line informing how many measurements were performed is not appropriate to this item as well as Table 2. This information should be described in “results”. Was the normality test performed? 

Answer: The sentence and Table 2 have been moved to the results section. A normality test was performed but not included in the manuscript. 

Results 

1) If the reference toothpastes did not promote enamel wear, would it be possible that the number of strokes was not enough and then the model should be reviewed? 

Answer: While the reference kinds of toothpaste did not promote significant enamel wear in our study, we believe that the number of strokes used was appropriate for the conditions simulated. We selected 5,000 strokes based on established protocols in similar studies within the literature, which have demonstrated sufficient abrasion to evaluate wear rates effectively. However, we acknowledge that other factors, such as the abrasiveness of the dentifrices and the simulation setup, could influence the wear outcomes. We will consider your suggestion and explore the potential need for model adjustments in future research to ensure a comprehensive evaluation of toothpaste efficacy in promoting enamel wear.

---

## [Decision Letter · Decision Letter 1]

29 Oct 2024

Relative Abrasive Potential of Silica-Based Conventional and Tablet Dentifrices on Enamel and Dentin

PONE-D-24-31778R1

Dear Dr. Tribst,

We’re pleased to inform you that your manuscript has been judged scientifically suitable for publication and will be formally accepted for publication once it meets all outstanding technical requirements.

Kind regards,

Geelsu Hwang, Ph.D.

Academic Editor

PLOS ONE

Additional Editor Comments (optional):

Reviewers' comments:

Reviewer's Responses to Questions

**Comments to the Author**

1. If the authors have adequately addressed your comments raised in a previous round of review and you feel that this manuscript is now acceptable for publication, you may indicate that here to bypass the “Comments to the Author” section, enter your conflict of interest statement in the “Confidential to Editor” section, and submit your "Accept" recommendation.

Reviewer #2: All comments have been addressed

Reviewer #3: All comments have been addressed

2. Is the manuscript technically sound, and do the data support the conclusions?

Reviewer #2: Yes

Reviewer #3: Yes

3. Has the statistical analysis been performed appropriately and rigorously? 

Reviewer #2: Yes

Reviewer #3: Yes

4. Have the authors made all data underlying the findings in their manuscript fully available?

Reviewer #2: Yes

Reviewer #3: Yes

5. Is the manuscript presented in an intelligible fashion and written in standard English?

Reviewer #2: Yes

Reviewer #3: Yes

6. Review Comments to the Author

Reviewer #2: The manuscript was improved after the review. All my comments were addressed. The title and other manuscript parts are better reflecting the work. The statistical analysis was improved.

Reviewer #3: (No Response)

7. PLOS authors have the option to publish the peer review history of their article (what does this mean?). If published, this will include your full peer review and any attached files.

Reviewer #2: No

Reviewer #3: No

---

## [Editor Report · Acceptance letter]

25 Nov 2024

PONE-D-24-31778R1 

PLOS ONE

Dear Dr. Tribst, 

I'm pleased to inform you that your manuscript has been deemed suitable for publication in PLOS ONE. Congratulations! Your manuscript is now being handed over to our production team.

Kind regards, 

on behalf of

Dr. Geelsu Hwang 

Academic Editor

PLOS ONE